# The Autophagy Protein ATG16L1 Is Required for Sindbis Virus-Induced eIF2α Phosphorylation and Stress Granule Formation

**DOI:** 10.3390/v12010039

**Published:** 2019-12-29

**Authors:** Matthew Jefferson, Benjamin Bone, Jasmine L. Buck, Penny P. Powell

**Affiliations:** 1Biomedical Research Centre, Norwich Medical School, University of East Anglia, Norwich Research Park, Norwich NR4 7TJ, UK; M.Jefferson@uea.ac.uk (M.J.); B.Bone@uea.ac.uk (B.B.); 2Virology, Ashford and St Peter’s Hospitals NHS Foundation Trust, Chertsey, Surrey KT16 0PZ, UK; jasmine.buck@nhs.net

**Keywords:** Sindbis virus, eIF2α, autophagy, stress granule, interferon, RNA binding proteins

## Abstract

Sindbis virus (SINV) infection induces eIF2α phosphorylation, which leads to stress granule (SG) assembly. SINV infection also stimulates autophagy, which has an important role in controlling the innate immune response. The importance of autophagy to virus-induced translation arrest is not well understood. In this study, we show that the autophagy protein ATG16L1 not only regulates eIF2α phosphorylation and the translation of viral and antiviral proteins, but also controls SG assembly. Early in infection (2hpi), capsids were recruited by host factors Cytotoxic Granule-Associated RNA Binding Protein (TIA1), Y-box binding protein 1 (YBX1), and vasolin-containing protein 1 (VCP), to a single perinuclear body, which co-localized with the viral pattern recognition sensors, double stranded RNA-activated protein-kinase R (PKR) and RIG-I. By 6hpi, there was increased eIF2α phosphorylation and viral protein synthesis. However, in cells lacking the autophagy protein ATG16L1, SG assembly was inhibited and capsid remained in numerous small foci in the cytoplasm containing YBX1, TIA1 with RIG-I, and these persisted for over 8hpi. In the absence of ATG16L1, there was little phosphorylation of eIF2α and low levels of viral protein synthesis. Compared to wild type cells, there was potentiated interferon protein and interferon-stimulated gene (ISG) mRNA expression. These results show that ATG16L1 is required for maximum eIF2α phosphorylation, proper SG assembly into a single perinuclear focus, and for attenuating the innate immune response. Therefore, this study shows that, in the case of SINV, ATG16L1 is pro-viral, required for SG assembly and virus replication.

## 1. Introduction

When RNA viruses infect mammalian cells, innate immune pathways including autophagy, apoptosis and the interferon signaling are induced to eliminate the virus [1]. In addition, some viruses activate PKR, leading to phosphorylation of eukaryotic initiation factor 2α (eIF2α). This halts host translation and leads to the formation of SG [2]. SG are non-membranous cytoplasmic ribonucleoprotein aggregates, typically containing mRNA and a stable core of RNA binding proteins, with a dynamic outer shell with diverse protein components, depending on the stress inducer [3]. During replication, viral double-stranded RNA (dsRNA) intermediates are recognized by pattern recognition receptors leading to the production of anti-viral cytokines such as Type 1 interferon [4]. Pattern recognition receptors include Toll-like receptor 3 (TLR3), localized in early endosomes, and the cytoplasmic RIG-like receptors (RLRs), RIG-I and MDA5, and PKR [5]. PKR, RIG-I and MDA-5 localize to SG following infection, together with other anti-viral proteins encoded by interferon sensitive genes (ISGs) such as PARP, ZAP and RNAi-binding ISGs [2,6]. In some cases, virus-induced SG are essential for efficient induction of Type I interferon. SG themselves may be either pro- or anti-viral, as they may sequester either factors needed for viral replication or innate factors required for anti-viral defenses [7]. 

Autophagy is another important innate response to virus infection. Autophagy is a stress response, which is activated to sequester pathogens within double-membrane vesicles called autophagosomes, to deliver them for degradation in lysosomes. Autophagy is activated during alphavirus infection [8]. Alphaviruses are a genus of the Togavirus family, and include Sindbis virus (SINV), Semliki Forest Virus (SFV) and chikugunya virus (CHIKV). They infect vertebrates and insects. They have a single-stranded, positive -sense RNA genome of approximately 12 kb with two open reading frames (ORF). The subgenomic RNA is translated into structural proteins, the capsid and envelope glycoproteins E1 and E2. The other ORF encodes the non-structural proteins nsp 1–4 [9]. Previous work has shown that autophagosomes accumulate during SFV infection due to the inhibition of autophagosome maturation by the viral glycoproteins [10]. A further study showed that SINV capsid protein binds to the autophagy cargo protein p62, an adaptor protein that targets it to the autophagosome [11]. In vitro, viral replication was independent of the autophagy protein ATG5, although in vivo, mice with disrupted neuronal Atg5 had a reduced clearance of viral proteins and increased accumulation of p62 aggregates in neurons [11]. Autophagy has been shown to enhance the replication of some viruses, such as picornaviruses and flaviviruses [12]. 

In this study, we investigated the effect of the autophagy protein ATG16L1 on eIF2α phosphorylation, early SG formation and modulation of viral and host protein synthesis. We showed that in wild type MEFs, SINV efficiently activated autophagy and eIF2α phosphorylation, with high levels of viral protein translation. The viral capsid protein redistributed to a single perinuclear SG comprising host RNA-binding proteins YBX1, TIA1, VCP, in addition to PKR and RIG-I. In contrast, in ATG16L1-/- cells, there was absence of eIF2α phosphorylation, and low viral protein synthesis. Host RNA-binding proteins failed to coalesce into a perinuclear SG, but were found co-localized with viral capsid in numerous small cytoplasmic puncta. Interferon synthesis and induction of ISG transcription were potentiated in ATG16L1 deficient cells, and this demonstrated that both SG assembly and autophagy were not required for interferon signaling. The results indicate that the autophagy protein ATG16L1 is important for the early phosphorylation of eIF2α and the switch from host to virus protein production. ATG16L1 is also involved in attenuation of interferon responses and the formation of perinuclear SGs after viral infection.

## 2. Materials and Methods 

### 2.1. Cells and Viruses 

Mouse embryonic fibroblast cells (MEF) and BHK cells were grown in Dulbecco’s Modified Eagles Media (DMEM) with GlutaMax ( Gibco, ThermoFisher Scientific, UK) supplemented with 10% fetal calf serum (FCS) and 1% Penicillin/Streptomycin. MEFs were isolated from E13.5 embryos and immortalized by serial passaging. All mouse studies followed appropriate animal use regulation and protocols at UEA. ATG16L1 -/- MEF cells were generated from ATG16L1fl/fl mice crossed with Rosa26-LacZ mice [13]. The *Atg16L1* gene was deleted with Adenovirus expressing Cre recombinase and isolated by FACS through activation of Lac Z expression after deletion of upstream stop sequences by the Cre recombinase. Matched parental control MEF cells were made from *Atg16 fl/fl* embryos without treatment with Cre recombinase. Immunostaining of ATG16L1 was used to assess absence of ATG16L1 protein, and staining of LC3 and WIPI used to show absence of autophagosomes [13]. SINV laboratory strain AR339 was from John Fazakerley, The Pirbright Institute. SINV mCherry.capsid virus was made from an infectious clone by in vitro transcription of the cDNA for dsTE12Q with mCherry fused to the *n*-terminus the capsid protein and a kind gift from B. Levine, University of Texas Southwestern Medical Centre [9]. cDNA was linearised with Xho 1, the RNA was transcribed using SP6 RNA polymerase and transfected into baby hamster kidney (BHK) cells. The supernatant containing virus was collected after 3 days, and stored at −80 ℃ Cells were incubated with virus diluted in media to moi 2–5 for one hour a before washing with fresh media and incubating at 37 ℃. Cells were incubated for the times stated in individual experiments.

### 2.2. Antibodies for Western Blotting and Immunostaining 

Rabbit anti-SINV antibody (used at 1 in 200) was a gift from Sondra Schlesinger, Washington University St Louis. Other antibodies were as listed below. They were used at the dilutions indicated and they were incubated either for 2 h at room temperature, or at 4 ℃ overnight. Rabbit anti- YBX1 (Abcam#ab12148, 1 in 300); rabbit anti-VCP (Cell Signaling Technologies, London, UK #2648); mouse anti-G3BP1 (BD, 611126, 1 in 200); goat anti-TIA1 (Santa Cruz sc-1751, 1 in 200); rabbit anti-PKR (Abcam#ab32052); rabbit anti-RIG-I (Abcam#ab45428, 1 in 200); rabbit anti-dsRed (Clontech 632496, 1 in 1000); mouse anti-ATG16L1 (MBL M150–3, 1 in 1000); mouse anti-actin (Sigma-Aldrich, Gillingham, Dorset, A5441, 1 in 7000); rabbit anti LC3 A/B (Cell Signalling#4108 1 in 1000). The antibody against phosphorylated eIF2 alpha was from Cell Signalling (#3398, 1 in 1000). Poly I:C (Roche) was transfected into cells with lipofectamine (Invitrogen) for four hours. Procarta mouse interferon alpha/interferon beta Luminex 2-plex was from Affimetrix (eBiosciences, ThermoFisher Scientific, UK). 

### 2.3. Immunoblotting

Protein was extracted with M-PER (Pierce, ThermoScientific, UK) for 30 min on ice in the presence of HALT protease inhibitor cocktail (Pierce). For phosphorylated proteins, cells were lysed in 1% hot SDS in Tris ph8.0 with phosSTOP phosphatase inhibitors (Roche) Proteins were separated by SDS-PAGE (4–12% acrylamide) and transferred to PVDF membranes (0.45um transfer membrane, ThermoFisher Scientific). Membranes were blocked with 5% (*w/v*) dried skimmed milk in TBS containing 0.5% Tween 20. Membranes were probed with anti-SINV, LC3 and actin antibodies as described. Proteins were detected with IRDye^®^-labeled secondary antibodies (Li-Cor Biosciences, Cambridge UK) at 1:10,000 dilution Proteins detected by the labelled secondary antibodies were visualized on the Odyssey infrared system

### 2.4. Immunocytochemistry and Fluorescent Microscopy

MEF cells and pericytes grown on coverslips were infected with SINV moi of 5 for the stated times and fixed in 4% paraformaldehyde for 10 min. Cells were blocked in goat serum in gelatin quench and permeabilized with 0.2% Triton X-100. Primary antibodies were diluted to the concentrations listed above into 0.2% Triton X-100 in goat gelatin quench and incubated with cells for 1 h at room temperature and cells were washed three times 0.1% Tween before incubation with fluorescent secondary antibodies. Images were analyzed using the Axioplan software version 4.8. For pixel density analysis, images were analyzed using Imaris ×64 7.2.3 software (Bitplane Software Incorporated, Oxford, UK).

### 2.5. Reverse Transcription PCR 

Total cellular RNA was isolated with Trizol. cDNA was generated with Superscript II and specific primers used for PCR were ISG-15: GCCCACCAAACTGCAGTGCTC and CTGCTGGGGGAGTATGGCCTAAAG; beta actin: GAGGCCCCCCTGAACCCTAAG and GAACCGCTCGTTGCCAATAG; IFN β:GTGGATCCTCCACGCTGCGTTC and TCTCTGCTCGGACCACCATCCA; MDA-5: CCCCGAGCCAGAACTGCAGC and ATCTGCGGCAGGGGAATGGC; RIG-I: CGTTGGGCTGACTGCCTCCG and TGCAGACCCGGCTCTCCTCC; PKR: CGAAAACTGCCGGAACATCC and CTCCACTCCGGTCACGATTTG; ISG1: GCCCACCAAACTGCAGTGCTC and CTTTAGGCCATACTCCCCCAGCAG; ISG56: ACAGCTACCACCTTTACAGC and TTAACGTCACAGAGGTGAGC; IP-10:TCATCCTGCTGGGTCTGAGT and CTGGGTAAAGGGGAGTGATG.

RNA was purified with Qiagen RNeasy MinElute kit for qPCR with Bioline Immomix™, SYBR green and Quantitech 10X primers 18S, IFNβ1, Isg15 or IFIT1 (Qiagen, Manchester, UK). It was performed on an ABI7500 real-time PCR machine and transcript levels were normalized relative to 18S using the ΔΔCt method.

### 2.6. Statistical Analysis

The data was analyzed by an unpaired student t test using Graphpad Prism 8 (www.graphpad.com/scientific-software/prism)

## 3. Results

### 3.1. ATG16L1 is Required for SINV-Induced Phosphorylation of eIF2α and the Switch to Viral Protein Synthesis

To investigate the effect of ATG16L1 on virus protein expression and eIF2α phoshorylation, wild type and ATG16L1-deficient MEF cells [13] were infected with SINV at an moi of 2. A time course of viral protein expression from 0–8 hpi using rabbit anti-SINV antibody showed that viral protein translation commenced by 6 hpi in both cell lines (Figure 1a). The enveloped glycoprotein p62 (E2/3) together with the capsid protein fused to mCherry fluorescent protein (ch.capsid), is seen as the major band at 62 kDa. The envelope protein E1 is detected at 45kDa and cleaved capsid protein at 30kDa. This anti-SINV antibody detected a non-specific, cross reacting band at this molecular weight at 0 hpi. The p62/capsid proteins increased 12-fold by 8 hpi in wild type cells. In contrast, there was only a 2-fold increase in p62/capsid proteins in ATG16L1 knockout cells (Figure 1a, bar chart). In wild type cells, SINV infection triggered phosphorylation of elongation factor eIF2α by 6 hpi. However, there was little phosphorylation of eIF2α detected by 6hpi in ATG16L1 knockout cells (Figure 1b). On this blot SINV virus expressing mCherry capsid is shown at 62kDa in wild type cells using an anti-dsRed antibody. There is a low intensity caspid band at 62kDa in ATG16L1 knockout cells, confirming the lower viral protein translation in these cells. The graph below shows the levels of eIF2α phosphorylation from three independent blots (+/-SE) at each time point, with 3-fold more eIF2α phosphorylated protein in wild type compared to ATG16L1 knockout cells. These results suggest that ATG16L1 is required both for eIF2α phosphorylation and for viral protein synthesis. 

### 3.2. Host Protein Rearrangements Following Entry of SINV Capsids 

Canonical SG are induced by sodium arsenate and they are comprised of host proteins TIA1, G3BP1and YBX1. In MEF cells, these RNA-binding proteins redistributed from the cytoplasm and nucleus to several perinuclear bodies after treatment with sodium arsenate (Figure 2a). In contrast, SG formed after infection with SINV wild type strain AR339 condensed into a single, large perinuclear granule, which also localised with host RNA binding proteins YBX1 and TIA1 (Figure 2b). As such, these granules are defined as canonical SG. Host proteins redistributed from the nucleus and cytoplasm and formed SGs by 4–8 hpi. The SG formed after virus infection was seen as a single perinuclear body. Similarly, during infection of the recombinant SINV mCherry.capsid, the virus capsid redistributed with YBX1 and TIA1 and also with VCP and G3BP1 into a single perinuclear granule (Figure 2c). 

RNA-binding proteins VCP, YBX1 and TIA1 were initially found in the nucleus and cytoplasm in both WT and ATG16L1 -/- cells (Figure 3a). The capsid redistributed with the RNA-binding proteins as early as 2hpi into small cytoplasmic puncta which coalesced into a single large perinuclear body in WT cells by 8 hpi (Figure 3b, WT). In ATG16L1 -/- cells, characterised in detail in Rai et al. [13], capsid redistributed with YBX1, TIA1 and VCP in the cytoplasm by 2 hpi, however a large perinuclear granule was not formed by 8hpi (Figure 3 b, ATG16L1-/-). In ATG16L1-/- cells, host RNA-binding factors and capsid were seen as numerous small puncta throughout the cytoplasm. When the number of cells having either a large capsid-containing perinuclear SG in wild type cells, or small cytoplasmic puncta containing capsid in ATG16L1-/- were quantitated, there was a significant difference in large and small granules between both cell types. However, the same percentage of cells were infected in both cell types (Figure 3c). Therefore, although the initial infection was the same in both cell types, viral protein synthesis shown in Figure 1a was higher in wild type cells. In wild type cells, SG formation does not halt translation of viral proteins. In contrast, in ATG16L1 knockout cells, viral protein synthesis was restricted, despite low levels of eIF2α phosphorylation and absence of perinuclear SG formation. These experiments followed labelled capsid at early time points before the start of virus replication. We detected dsRNA with antibody J2 in separate replication complexes. 

Inhibition of degradation by autophagy may explain the accumulation of the many small cytoplasmic puncta containing capsid and RNA-binding proteins seen in ATG16L1-/- cells. To investigate whether autophagy affects turnover of these host translation factors, we investigated protein levels during activation of autophagy by SINV in wild type cells (Figure 4). 

During autophagy, cytoplasmic LC3 (LC3I) is conjugated to phosphatidylethanolamine to form membrane-bound LC3II, by proteins, including ATG16L1, that are part of an ubqitination-like conjugation system. By 4 hpi, LC3I had been converted to the faster migrating LC3II form, demonstrating that SINV indeed activates autophagy early in infection. In ATG16LI -/- cells, LC3I was not converted to LC3 II during infection, confirming that these cells lack autophagy and also demonstrating that the puncta in Figure 3b are not autophagosomes There was no change in levels of YBX1, VCP and TIA1 over the time course of 0 to 8 hpi in wild type cells, indicating that they are not degraded by autophagy during this time period. In ATG16L1-/-, protein levels for TIA1, YBX1 and VCP also remained constant. Therefore, inhibition of autophagy does not lead to the accumulation of these host translation proteins, and the small cytoplasmic puncta seen in ATG16L1 knockout cells are not due to block in degradation of these factors. 

### 3.3. Innate Responses to SINV in Autophagy-Deficient Cells

We examined the localisation of dsRNA viral sensors PKR and RIG-I. In wild type cells, by 2 hpi, both PKR and RIG-I had relocated from the cytoplasm (Figure 5a, CON) to a single capsid-containing perinuclear SG (Figure 5b WT), which remained for up to 8hpi. In ATG16L-/- cells, PKR and RIG-I also redistributed with capsid but there were smaller granules, less tightly packed and which became dispersed throughout the cytoplasm by 8hpi (Figure 5b, ATG16L1-/-). These small cytoplasmic puncta at 8hpi contained SINV capsid, RIG-I and TIA1 (Figure 5c), marking them as canonical SG.

Interferon synthesis is required to induce several interferon stimulated genes or ISGs [14]. Type 1 interferon secretion was measured after SINV infection of wild type and ATG16L1 -/- cells for 24 hpi, using a sensitive Affymetrics Luminex IFN β assay kit. Figure 6a shows the fold increase in interferon secreted from ATG16L1-/- treated with poly IC, or infected with SINV, compared to wild type cells. We found that the assay varied with different virus preparations; therefore we treated each experiment independently for 5 biological replicates using the same virus stock in each experiment. Both dsRNA (poly I:C) treatment and SINV infection increased interferon secretion in ATG16L1 knockout cells by 1- to 4-fold over wild type cells. To examine if this was due to early induction of interferon synthesis, we investigated ISG mRNA expression, using total RNA isolated from SINV infected cells at 0, 4, 6, 8 and 24 hpi. Transcription of ISGs, MDA-5, RIG-I, PKR, ISG-15, ISG-56 and IP-10 were analysed by semi-quantitative RT-PCR with specific primers (Figure 6b). SINV infection led to a rapid increase by 4hpi in mRNA for ISGs RIG-I, PKR, MDA-5, with the greatest increase being ISG15 and ISG56 mRNA, which were undetectable before infection. This demonstrates that ISGs were induced through the production of interferon as early as 4hpi. We used IRF3 mRNA levels to represent a non-responsive gene.

Real-time quantitative RT-PCR was used to measure relative differences between wild type and ATG16L1 knockout cells for IFN beta, ISG15 and IFIT1 (ISG56). When cells were treated with poly I:C for 4 h, there was a small significant difference (*p*-value < 0.5) between wild type and ATG16L1 knockout cells in interferon beta mRNA expression levels, but no difference for ISG15 or IFIT1 mRNA (Figure 6c). However, when cells were infected with SINV for 4 h, there was 100-fold greater IFN beta, ISG15 and ISG56 mRNA detected in cells lacking ATG16L1 compared to wild type cells (Figure 6c). In summary, interferon translation occurs in wild type cells in the presence of SG, at a time when eIF2a is phosphorylated and translation factors had been sequestered into SG during virus infection. However, there is a potentiation in interferon production and ISG transcription in ATG16L1 knockout cells, when RNA-binding host translation factors are scattered throughout the cytoplasm, and when there was little eIF2α phosphorylation and low viral protein synthesis.

## 4. Discussion

Here we show that, following SINV infection, ATG16L1 is required for the efficient phosphorylation of eIF2α, for high levels of viral protein synthesis and for the formation of the single perinuclear SG containing capsid seen early in infection. Generally, eIF2α phosphorylation leads to host translational arrest. It can be phosphorylated by over four different kinases, PERK, PKR, heme-regulated eIF2α kinase (HRI), and general control nonderepressible 2 (GCN2) [15]. PKR is activated mainly by dsRNA during viral infection, although GCN2 can also be activated by SINV RNA genomes [16]. For most viruses, this would also block viral replication. However, it does not affect the translation of alphavirus proteins. SINV can promote its own translation in the presence of phosphorylated eIF2α, through a translational enhancer, which is a stable hairpin loop in the 26S subgenomic promoter that stalls the ribosome on the initiation codon to enhance translation of the structural proteins [17,18]. In our study, when there was a diminished eIF2α phosphorylation in ATG16L1 deficient cell, there was a lower viral protein translation. It demonstrates that the translation enhancer requires eIF2α to be phosphorylated for virus translation to begin.

SINV infection promotes both stress and autophagy [19]. Autophagy and translation are inversely regulated, and previous work has shown that phosphorylation of eIF2α is essential for both starvation-induced and virus-induced autophagy [20]. Cells with a non-phosphorylatable mutant of eIF2α (S51A) did not induce autophagy in response to starvation [21]. We show that that autophagy protein ATG16L1 is essential for eIF2α phosphorylation and to initiate SG formation. ATG16L1 has a well-defined role in canonical and non- canonical autophagy, but it is unclear how it regulates PKR. Autophagy is initiated when stresses such as starvation and infection increase AMPK activity, which inactivates the mechanistic target of rapamycin complex (mTOR). Autophagy proteins in the ULK1 multi-protein complex phosphorylate phosphatidylinositol 3-phosphate (PI3P) to act in autophagosome nucleation. ATG16L1 interacts with ATG12-ATG5 to mediate the conjugation of LC3 to phosphatidylethanolamine (PE) on autophagosomes, leading to membrane elongation and expansion. However, this occurs in late infection, after 8hpi. ATG16L1 is also associated with non-canonical autophagy, the phagocytosis of extracellular pathogens into single membrane vesicles in a process known as LC3 associated phagocytosis (LAP). Roles of ATG16L1 in other processes, such as endocytosis, exocytosis and endosomes/lysosome repair have been described [22,23]. This study highlights another role for ATG16L1 in eIF2α kinase signaling.

SGs acted as platforms for sensing viral dsRNA, as RLR sensors, PKR and RIG-I localized to virus-induced SG very early in infection [24]. In wild type cells, RNA binding proteins TIA1, YBX1 and VCP redistributed with PKR and RIG-I into a single perinuclear body containing SINV capsid. However, in cells deficient in ATG16L1, RIG-I and PKR were scattered with TIA-1 and capsid throughout the cytoplasm, and did not coalesce into a single SG. The accumulation of cytoplasmic granules was not due to inhibition of autophagy, as we showed that host RNA binding proteins were not degraded by autophagy in wild type cells. In addition, it suggests that SG assembly is not an important anti-viral defense against SINV, as perinuclear SG formation yielded higher virus protein synthesis in wild type cells, and they were dispensable for interferon production in ATG16L1 knockout cells. The small cytoplasmic puncta containing RIG-I in ATG16L1-/- cells correlated with increased interferon and early ISG induction. The fact that ATG16L1 is important in initiating virus protein translation, and important for attenuating the anti-viral response could be linked. One mechanism by which autophagy can negatively regulate interferon signaling is through ATG5–ATG12 interacting with the CARD domain of RIG-I and MAVS to suppresses IFN induction [25]. Alphavirus can also prevent host recognition of its viral dsRNA [26] and it can inhibit synthesis of IFN and ISGs transcription and translation [27]. Interestingly, capsid proteins and nsp2 proteins can inhibit both transcription and translation [28]. In other work, SG formation by inducers such as proteasome inhibitors was inhibited when autophagy was inhibited [29].

The phosphorylation of eIF2α decreases translation of most mRNAs. However, it can increase translation of a selected number of mRNAs containing short open reading frames. We show using a sensitive ELISA that interferon translation is increased in SINV infected wild type cells compared to control cells, and this correlates with an increase in ISG mRNAs, where transcription is switched on by interferon. This transcriptional upregulation of host antiviral proteins in ATG16L1 -/- cells may help to decrease virus production. The synthesis of interferon is potentiated in ATG16L1-/- cells, at the same time as perinuclear SG formation is inhibited. This suggests that SG formation may not be required for interferon signaling, as previously suggested [30], and indeed lack of SG results in higher interferon translation. Further work using puromycin labelling of the various types of granule would determine which granule is most translationally active. 

Our work here adds to the complex story of SG assembly during alphavirus infection. It has been well established that SINV disassembles SG late in infection from 8–16 hpi [31]. Different types of complexes containing viral nsP3- with G3BP1 and YBX1 have been isolated from both insect and mammalian cells [32], and sequestration of G3BP1 into replication complexes disperses SGs [31]. Similarly, CHIKV induces G3BP1 and G3BP2 –containing granules containing nsp2 and nsp3, which are not classical SG, but needed for efficient virus replication [33,34]. Our study correlates the inhibition of assembly of SGs in the absence of ATG16L1, with potentiated interferon responses and indicates that it is not the assembly of SG that enhances anti-viral responses, but the lack of eIF2α phosphorylation that decreases viral protein synthesis. 

## Figures and Tables

**Figure 1 viruses-12-00039-f001:**
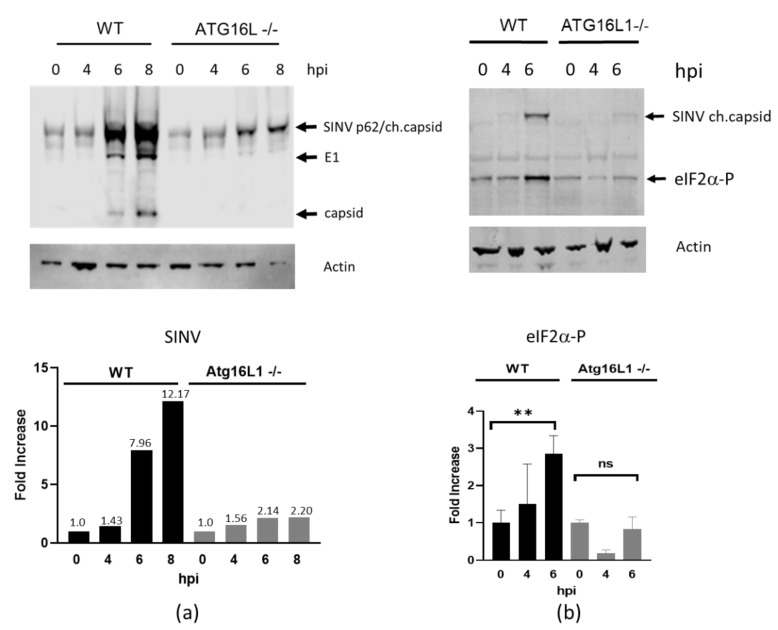
eIF2α phosphorylation and virus protein production is diminished in ATG16L1-/- cells. (**a**) Western blot of SINV-infected lysates at 0–8 hpi detected with anti-SINV antibody. Bar chart below shows fold increase in p62/capsid proteins relative to actin. (**b**) SINV expressing cherry capsid detected with anti-dsRred antibody and phosphorylated eIF2α detected with rabbit monoclonal phospho-eIF2α (Ser51). Bar chart below shows the fold increase in eIF2α-P normalised to actin for 3 independent blots with +/-SE ** *p* < 0.01, ns is non-significant.

**Figure 2 viruses-12-00039-f002:**
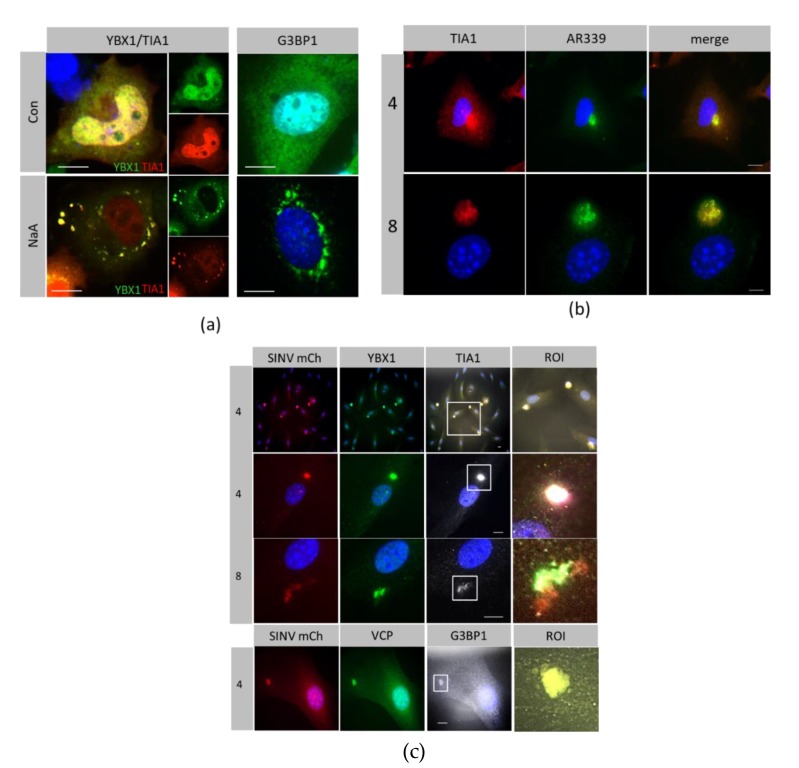
RNA binding proteins redistributed with SINV capsid early in infection. Host RNA-binding proteins YBX1, TIA 1, G3BP1 and VCP were detected by immunostaining (**a**) Control MEF cells (Con) and cells treated with 100 nM sodium arsenate for 2 h (NaA) (**b**) MEFs infected with SINV strain AR339 for 4 and 8 hpi (**c**) Cells infected with SINV Cherry.capsid for 4hpi (low power x40), 4 hpi (x63) and 8 hpi (x63). Region of interest (ROI) are from merged images of white squares. Scale bar = 10 μm.

**Figure 3 viruses-12-00039-f003:**
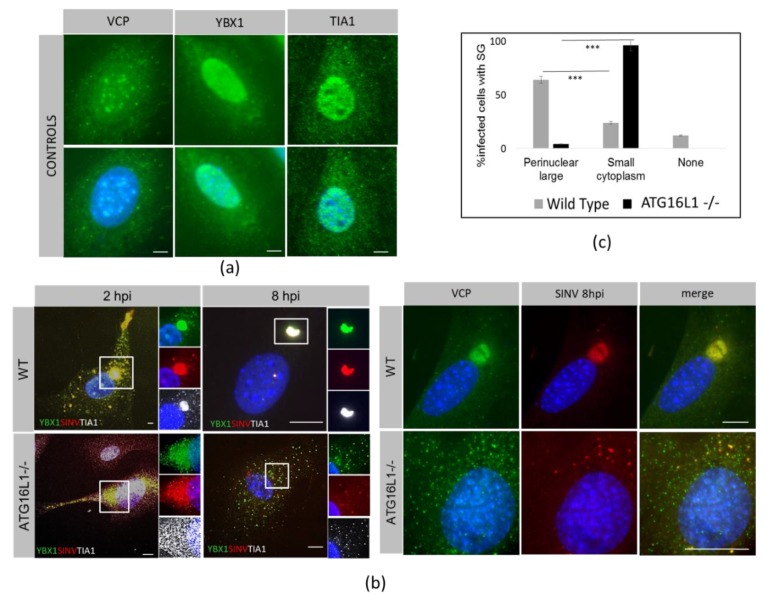
Perinuclear stress granules do not form in ATG16L1-/-cells.Host proteins YBX1, TIA1 and VCP were stained in WT and ATG16L1-/- MEF cells. (**a**) distribution in ATG16L1 uninfected cells. (**b**) distribution of host proteins in WT and ATG16L1 MEFs cells infected with SINV mCh.capsid for 2 and 8 hpi. (**c**) The bar chart shows percentage of infected cells with either large perinuclear SG or small cytoplasmic puncta from 5 independent fields of view of WT and ATG16L1-/- cells at 8hpi at 40× magnification. +/- SE and ****p* < 0.001. None indicates a small number of infected cells with no SG.

**Figure 4 viruses-12-00039-f004:**
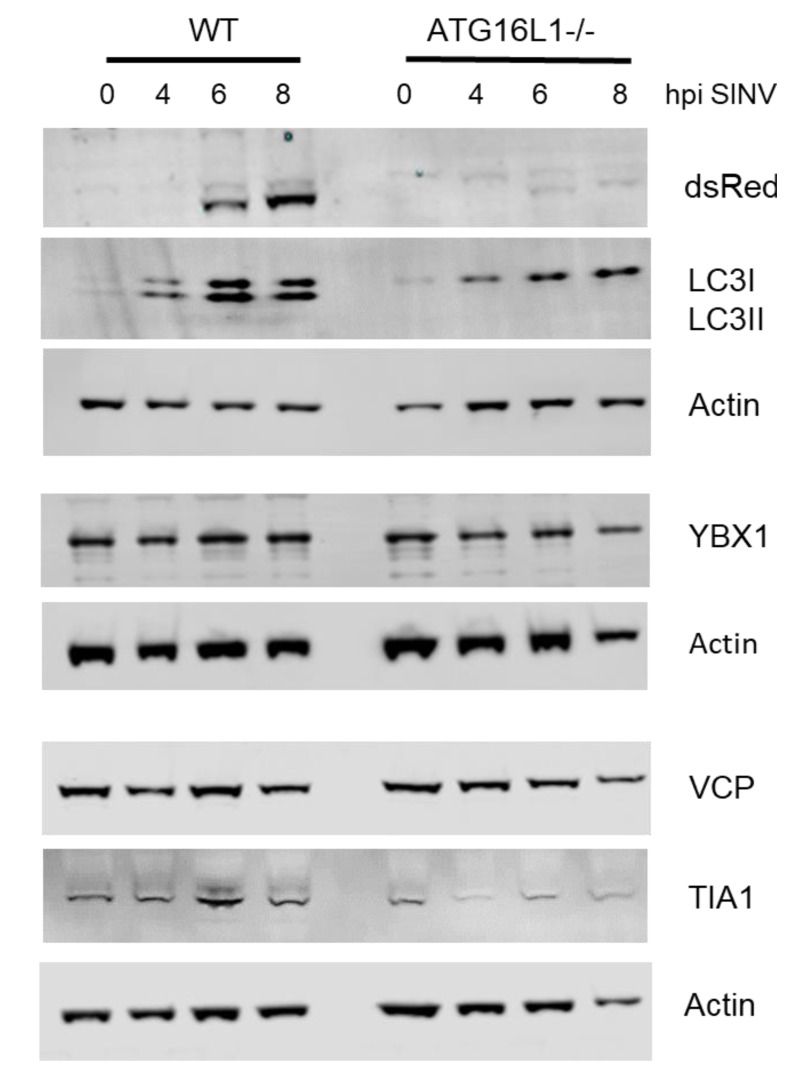
Host translational RNA-binding proteins are not degraded by autophagy induced by SINV infection. A time course of SINV infection from 0–8 hpi in wild type (WT) and ATG16L1-/- MEFs (ATG16L1-/-). Western blot of protein lysates with antibodies against LC3, YBX1, TIA1 and VCP, with actin showing equal lane loading for each gel.

**Figure 5 viruses-12-00039-f005:**
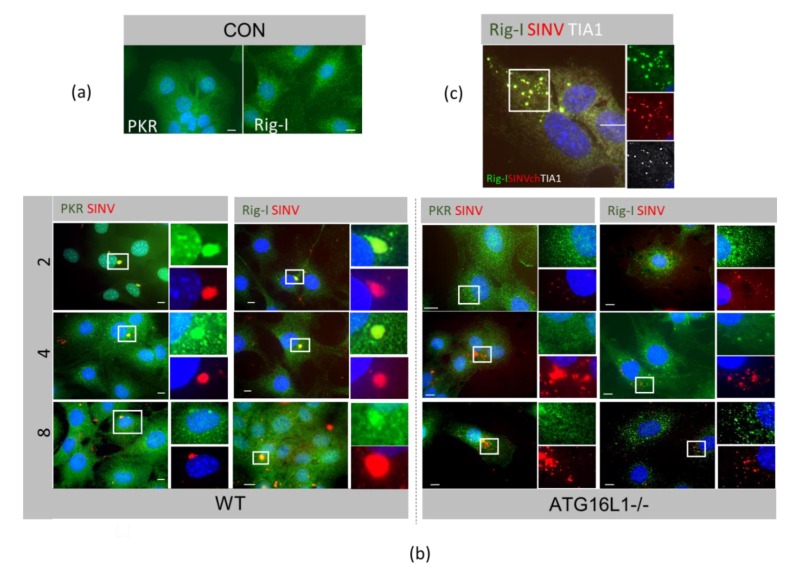
Viral dsRNA sensors PKR and RIG-I redistribute with viral capsid. Cells were immunostained for PKR, RIG-I and TIA-1. (**a**) Control uninfected wild type MEFs (CON) (**b**) Wild type (WT) and ATG16L1-/- cells infected with SINV mCh.capsid for a time course of 2, 4 and 8 hpi. (**c**) ATG16L1-/- infected with SINVmCh.capsid for 8 hpi.

**Figure 6 viruses-12-00039-f006:**
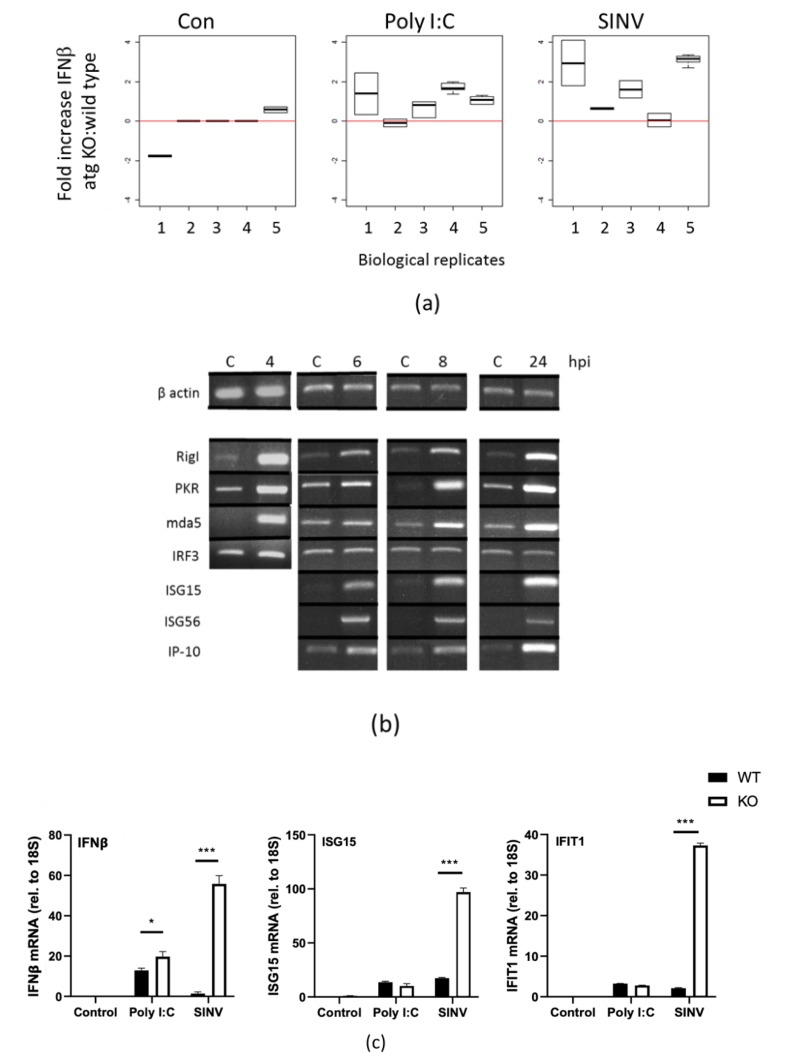
ATG16L1 attenuates interferon β mRNA and protein secretion, and ISG gene expression in SINV infected cells.(**a**) Interferon β secretion from control cells cells (CON), cells treated with synthetic dsRNA (Poly I:C), or cells infected with SINV. Fold increase in interferon secretion from ATG16L1-/- cells relative to wild type cells is shown for control cells, cells transfected with poly I:C with lipofectamine and cells infected with SINV for 24 hpi. Graph shows values for 5 independent biological replicates. (**b**) MEF cells were infected with SINV for 4, 6, 8 and 24 hpi with matching controls isolated at the same time point. mRNA for ISGs RIG-I, PKR, MDA5, ISG15, ISG56 and IP-10 are shown by semi-quanitative RT-PCR. Beta actin shows equal lane loading and IRF3 is a non-responsive gene. (**c**) qPCR using primers for IFNβ, ISG15 and IFIT1 (ISG56) using total RNA from MEFs (WT) and MEFs deleted in ATG16 (ATG16L1-/-) either transfected with dsRNA (Poly I:C) or infected with SINV for 4 h. WT cells (black bars) and ATG16L1 -/- cells (white bars). ΔΔCt relative to 18S. (*) *p*-values < 0.05; (***) *p*-value < 0.001.

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
