# Peer review of "The Autophagy Protein ATG16L1 Is Required for Sindbis Virus-Induced eIF2α Phosphorylation and Stress Granule Formation"

_viruses, 2019, doi:10.3390/v12010039_

Round 1
Reviewer 1 Report
This paper evaluates role of the autophagy protein ATG16L1 in eIF1-alpha phosphorylation and stress granule formation during the early period following Sindbis virus infection. While the subject is of interest to the virology field and the general approaches to answering the presented questions were sound, the manuscript was extremely difficult to read and interpret due to poor quality of presentation. The text contains multiple grammatical and punctuation errors and has numerous run-on sentences. In multiple paragraphs, the sentences appear to be randomly distributed, providing no flow of thought and making it very challenging for the reader to follow the story being laid out by the authors. The figures lack several needed control graphs and panels, and the figure legends do not provide enough information, such as sample size or statistical analyses, to adequately interpret the data provided in the figures. The Materials and Methods do not provide enough detail for the reader to determine how experiments were performed. Together, these shortcomings make it very difficult to adequately interpret the scientific soundness of the data being presented.
Specific Comments:
The Introduction is difficult to follow and should be critically reviewed for clarity and flow. Sentences within the same paragraph appear to be inserted randomly, with no flow or thought continuation. The lack of commas separating thoughts in several sentences is resulting in run-on sentences in multiple places.
Line 34 - In the sentence "...the translation of virus proteins is not affected..", by what is SINV translation not affected? Phosphorylation by eIF2-alpha?
Lines 39-47 - The way in which stress granules are introduced in the first paragraph is very confusing; the final sentence in the paragraph (lines 44-47) should be moved farther to the front of the paragraph.
Lines 48-50 - The sentence switches verb tenses midway through and should be changed so that the verb tense is consistent throughout not only the sentence, but the full paragraph.
Lines 50-53 - This sentence should be moved to the subsequent paragraph, as it appears out of place here.
Lines 57-73 - This paragraph should be shortened to just include the study question and the major relevant findings.
Lines 76-80 - Please add a sentence confirming that all mouse studies followed appropriate animal use regulations and protocols at the research institution.
Materials and Methods - Throughout the section, the authors make several references "...as described" in place of an actual description of the of procedures conducted but provide no citation for the reader to reference. Please expand this section so that a reader may be able to conduct the same experiment based off of the description provided.
Lines 75-90 - Please provide a more detailed description of how cells were infected with SINV (MOI, length of virus incubation, etc).
Materials and Methods - Please provide the dilutions and length of incubation for all antibodies used.
Materials and Methods - Please describe the statistical tests used to analyze the data.
Line 137 - What MOI was used to infect MEF cells with SINV?
Figure 1a - How was SINV capsid being detected by immunoblot at 0 HPI? This causes this reviewer to suspect SINV capsid is actually the smaller band visible at 6 and 8 hpi present immediately below the band indicated by the arrow as SINV capsid. How many independent blots are represented in this graph?
Results section 3.2 - What cells were used for these experiments?
Figure 2a - These images would be easier to interpret if the YBX1 and TIA1 channels were shown separately in addition to merged. As currently shown, this reviewer is not convinced that TIA1 is redistributing from the nucleus to perinuclear bodies. The control G3BP1 image appears to be of a different magnification than the NaA G3BP1 image and should be clarified in the figure legend. Why wasn't VCP staining shown in this figure panel?
Figure 2b - This figure would benefit from a 0 hpi panel showing the baseline staining. The image of the cell at 4 hpi appears to be a different magnification than the image of the cell at 8 hpi and should be clarified in the figure legend.
Figure 2c - This figure would benefit from a 0 hpi panel showing the baseline staining. Are the ROI panels merged images of all the channels, or just zoomed in versions of certain proteins (and if so, which ones)?
Figures 2, 3, and 5 legends - The microscopic procedure details (types of antibodies, etc) should be moved from the figure legend to the Materials and Methods.
Lines 225-239 - What is the distribution pattern of YBX1, TIA1, and VCP in ATG16L1 KO cells at baseline/prior to infection? Images of fluorescent staining in uninfected ATG16L1-/- cells should be added to Figure 3.
Lines 232-233 - Figure 3c does not show the percentage of cells infected in WT vs ATG16L1 KO cells, despite being stated as such in the text.
Lines 233-234 - Figure 3 does not show quantification of viral protein synthesis, but rather distribution, so this statement is not supported by the figure.
Figure 3 - Did the authors perform fluorescent staining of ATG16L1 to confirm full knockout of the protein from ATG16L1-/- cells? This should be shown.
Figure 3a and 3b - Why are these separate panels? The presentation of this information would be improved if the VCP staining were included with the staining of the other SG proteins in Fig 3a.
Figure 3c - The graph is rotated 90 degrees and difficult to read in this orientation. At what magnification were the 5 independent fields examined? What does the "none" category in the graph represent?
Line 288 - Please provide some background regarding LC3I to LC3II conversion in the context of autophagy for readers who are not in the autophagy field.
Figure 5b - The Rig-1/SINV merged channels at 2 and 4 hpi appear to be switched with the RIG-1 alone channel relative to the other panels in the figure. The SINV distribution at 2 hpi in the ATG16L1-/- cells appears more like a large perinuclear SG rather than numerous cytoplasmic puncta, which conflicts with the data presented earlier in the manuscript.
Figure 6a - Because of the variability of the results across biological replicates, this reviewer would not trust the results of this assay and recommend it be removed from the paper. Perhaps performing the assay on cell homogenates rather than supernatents would be provide more reliable data.
Figure 6b - A graphical representation of the intensity of the semi-quantitative RT-PCR bands presented in this figure would be helpful.
Lines 324-331 and 355-360 - Why was semi-quantitative RT-PCR used for some genes, while qRT-PCR was used for others?
Lines 356-358 - The text states that there was no difference between gene expression in WT vs ATG16L1 cells following poly I:C treatment, but according to Figure 6c, there was a significant difference between cells types for IFN-beta expression. Please reconcile.
Figure 6c - A legend indicating what the black and white columns represent is needed for these graphs.
Discussion - Similar to the Introduction section, this section contains numerous grammatical and punctuation errors and should be critically reviewed and corrected.
Author Response
We thank the reviewer for his comments. We hope we have addressed his concerns in the revised manuscript.
We have substantially re-written the Introduction and re-organised it in line with his specific suggestions. It has been critically reviewed for clarity and flow.Specifically: We have re-ordered paragraph 1 ( now lines 31-46), and introduced stress granules earlier in the paragraph.
We have added a fuller explanation of autophagy in paragraph 2, and introduced alphaviruses and their genomic organisation.
We have checked that the verb tenses are consistent throughout.
We have shortened the final paragraph to just include the study question and the major findings.
Materials and Methods.
Line 80: We have added that all mouse studies followed the appropriate regulations and protocols.
We have removed “as described” and replaced with the actual procedure
Line 93-94: We have added more detail of how cells were infected with SINV.
Lines 94-101 We have provided dilutions and length of incubation for all antibodies
Line 143: Statistical test has been added
Results.
Line 149: SINV moi has been added
Figure 1a: This is representative of at least four separate blots. There is a non-specific band picked up by the rabbit anti-whole virus antibody that we have consistently seen in uninfected cells. We have clarified the bands in this Western blot with additional labelling. In wild type cells, the larger band is the p62 glycoproteins, together with the fusion protein of mCherry.capsid ( 30kDa+32kDa). The smaller band at 45K Da is E1 glycoprotein, and the 30kDa band is cleaved capsid. This is consistant with published data using this antibody from Sondra Schlessinger lab. This is confirmed in Fig 1b, where the dsRed antibody, which only detects fluorescent cherry protein, shows a less intense band at 65kDa. We have added this explanation in lines 150-160.
Line 176: Added MEF cells
Figure 2a left panels. We have added staining of TIA1 and YBX1 in control uninfected cells at 0 hpi to show base line staining for Figures 2a,b and c. We used G3BP1, TIA1 and YBX1 as markers of canonical SG. VCP was not used as it does not thought to define canonical SG. The ROI are from merged images and this explanation has been added to the legend for Fig 2c.
Figures 2, 3, 5 The types of antibody and microscopy procedures have been removed and added to Materials and Methods.
Distribution patterns of YBX1, TIA1 and VCP1 in uninfected ATG16L1-/- cells has been added as Figure 3a.
Description of Figure 3c has been corrected Lines 199-202.
Line 202-203. The increase in virus protein synthesis is shown in Figure 1 a and this has been added to this sentence.
Line 196: Added ATG16L1 knockout cells have been extensively characterised in Rai et al ref [14]. Also, it is confirmed in Figure 4, where there is no conversion of LC3 I to LC3 ii after infection.
Figure 3a and b are now in the same panel (new Figure 3b)
Figure 3c The graph has been rotated by 90 degrees, the magnification for the independent fields and explanation of none have been added to legend.
Lines 225-230 provide a more detailed explanation of autophagy pathway and the LC3I conversion to LC3 ii
Figure 5b. The Rig-I/SINV merge has been corrected. We have chosen a clearer example of cytoplasmic puncta for RiG-I and PKR after SINV infection at 2 hpi.
Figure 6a. This figure indicates that translation of Type 1 interferon generally increases with ATG16L1-/- , despite disruption in SG formation. Its important that the presence of interferon is indicated because ISG mRNA expression is increased in Figure 6 b and c
Semi-quantitative RT-PCR was used for a wide panel of ISGs over a time course to choose the best times to isolate RNA for qPCR for maximum induction in WT versus ATG16L1-/- cells. A graphical representation of this data is not possible because the data are not linear, the bands in the infected lanes are saturated, but show that the ISGs are greatly induced.
Lines 281-283. We have corrected the significant difference for IFN beta following poly I:C treatment
Discussion.
The discussion has been significantly rewritten to correct punctuation and grammar.
Reviewer 2 Report
Jefferson and colleagues investigate the role of the autophagy protein ATG16L1 during Sindbis virus (SINV) infection of MEF cells. They demonstrate that ATG16L1 is required for SINV-induced eIF2α phosphorylation, stress granule (SG) formation and virus replication as well as for decreasing the interferon responses.
The question is very interesting and the results are of interest to the field. This is also a well written manuscript, and the experiments are well conducted.
I will only have minor comments that should be addressed by the authors.
1. In figure 2b, no SINV capsid was observed in ATG16L1 KO cells while it is indicated line 144 that SINV capsid was detected in both wild type and ATG16L1 knockout cells, and again … 2. Figure 1a, I am surprised by the fold increase in SINV capsid after 8 h of infection given the western blot. Quantification is required in the corresponding histogram.3. Line 168 : lysates instead of lystaes4. There is a problem in the numbering of the pages in different locations, as for the figure 1, the lines 312-316…5. Line 219, the SINV strain is the strain 339 and not 3996. Figures 2, 3 and 5 the size of the cells (nuclei) are very different. The authors should present cells with the same enlargement and indicate the scale bars.7. Lines 300-301, the sentence should be corrected8. Line 304, the fold increases indicated in the text are overestimated9. The authors should better explain the IRF3 control in the figure 610. In the legend of the figure 6, several points should be modified : treatment by poly I:C is not indicated, poly I:C is not transfected into cells, cells are not infected with SINV capsid, the significance of the bars (white and black bars) is not indicated… 11. Line 342, the authors indicate that there was no significant difference between wild type and ATG16L1 KO cells in mRNA expression levels for IFNgamma, ISG15 and IFIT1 after treatment with poly I:C but a significative difference is presented for IFNgamma mRNA level.12. The authors should homogenise the abbreviations used : controls, RIG-I…13. The authors should better discuss the role of specific ATGs versus the global autophagy mechanism in infections, and especially SINV infection. In the same way, the link between the eIF2a pathway and autophagy gene expresssion should be developed the discussion together with the corresponding references.
Author Response
We thank the revwiever for his helpful comments and we have made the following changes.
In Figure 1b (not 2b?), the sentence has been changed to “ SINV virus cherry capsid, detected on the same blot with dsRed antibody in Figure 1b, appeared by 6hpi in wild type cells and there was a faint band of the same size in ATG16L1 knockout cells.” Quantitation is now given on the histogram in Fig 1a 168 Lysates Problem in the numbering of the pages in different locations has been changed using the Layout Options choosing Top and bottom to remove the line numbers from Fig 1, 2, 3, 5 The SINV strain has been corrected to AR339 The scale bars have been added to Figures 2, 3, 5. The nuclei are different sizes, depending on whether we are indicating the dispersed puncta throughout the cytoplasm, or focusing on the single perinuclear stress granule. The scale bars should now highlight this. We have corrected the lines 300-301 to “Figure 6a shows the fold increase in interferon secreted from ATG16L1 knockout cells treated with poly IC or infected with SINV compared to wild type cells” In line 304 we have changed the fold increases to 1- and to 4-fold in ATG16L1 knockout cells over wild type cell. We have better explained the IRF3 control in Figure 6 “. We used IRF3 mRNA levels as a control to indicate a housekeeping gene that is not an ISG.” In the legend to Figure 6 we have added : Interferon beta secretion from control cells cells (CON), cells treated with synthetic dsRNA |(Poly I:C), or cells infected with SINV. We have added SINV and removed the name SINV mCherry.capsid to identify the virus.We have indicated that wild type cells are the white bars and ATG16L1 black bars and placed this in the legend in Fig 6 as well
11.We have changed line 342 “ When cells were treated with poly I:C for 4 hours, there was a small significant difference (p-value<0.5) between wild type and ATG16L1 knockout cells in interferon beta mRNA expression levels for these three genes, but no difference for ISG15 or IFIT1 mRNA (Figure 6c)”.
We have homogenised the abbreviations eg: RIG-I and Controls We have discuss the role of specific ATGs versus the global autophagy mechanism in infections, and especially SINV infection. In the same way, the link between the eIF2a pathway and autophagy gene expresssion should be developed the discussion together with the corresponding referencesRound 2
Reviewer 2 Report
The authors have made a commendable effort to address the issues raised in the first submission, and the last version answered all my questions… But there are still several minor issues:
Line 93, the “a” after one hour should be suppressed
Line 225, ubiquitination
Line 227, ATG16L1
Line 280, *: p<0.05 and not p<0.5
Line 282, the assertion: “there was100 fold greater IFN beta, ISG15 and ISG56 mRNA detected in cells lacking ATG16L1 compared to wild type cells (Figure 6c)” is an over interpretation of the data. This is only true for ISG15.